# Neuroexplicit Diffusion Models for Inpainting of Optical Flow Fields

## Abstract

Deep learning has revolutionized the field of computer vision by introducing large scale neural networks with millions of parameters. Training these networks requires massive datasets and leads to intransparent models that can fail to generalize. At the other extreme, models designed from partial differential equations (PDEs) embed specialized domain knowledge into mathematical equations and usually rely on few manually chosen hyperparameters. This makes them transparent by construction and if designed and calibrated carefully, they can generalize well to unseen scenarios. In this paper, we show how to bring model- and data-driven approaches together by combining the explicit PDE-based approaches with convolutional neural networks to obtain the best of both worlds. We illustrate a joint architecture for the task of inpainting optical flow fields and show that the combination of model- and data-driven modeling leads to an effective architecture. Our model outperforms both fully explicit and fully data-driven baselines in terms of reconstruction quality, robustness and amount of required training data. Averaging the endpoint error across different mask densities, our method outperforms the explicit baselines by $11 - 27\%$, the GAN baseline by $47\%$ and the Probabilisitic Diffusion baseline by $42\%$. Therewith, our method sets a new state of the art for inpainting of optical flow fields from random masks.

## 1 Introduction

Diffusion is a fundamental process in physics that leads to an equilibrium of local concentrations. It explains many phenomena and finds applications in image processing (Weickert, 1998) and computer vision tasks (Weickert & Schnörr, 2001). In particular, it motivates the smoothness term in dense optical flow estimation in variational techniques. However, recent advancements in optical flow estimation have been dominated by deep-learning approaches (Dosovitskiy et al., 2015; Ilg et al., 2017; Teed & Deng, 2020; Xu et al., 2022; Huang et al., 2022). While all these architectures include a task-specific model-driven operation that represents the data term, the regularization through the smoothness term is handled in a fully data-driven manner by the learned parameters of the convolutions (Dosovitskiy et al., 2015; Ilg et al., 2017; Teed & Deng, 2020) and more recently by attention (Xu et al., 2022; Huang et al., 2022). Notably, none of these approaches utilize a specialized regularization operation as it has been studied in traditional computer vision. For this reason, we investigate whether it is possible to integrate diffusion with its rich mathematical foundations based on partial differential equations (PDEs) into neural architectures. In the following, we refer to these model-driven operations that integrate specialized domain knowledge for a certain task as *explicit*, the general data-driven operations as *neural*, and the combination of both as *neuroexplicit*.

The role of the regularization in traditional computer vision is to propagate information from confident correspondences to regions with less or little information. Variational methods do so using smoothness terms that lead to diffusion terms in the Euler-Lagrange equations. To isolate this behavior, we focus on inpainting of sparsely masked optical flow fields and compare our novel architecture with popular state-of-the-art methods. By imposing the diffusion behavior explicitly, our goal is to achieve interpretable models with fewer parameters, improved generalization capabilities, and less dependence on large-scale datasets.

## 1.1 Contributions

For the first time, we implement an end-to-end trainable network to predict the diffusion tensor used in an image driven diffusion inpainting of an optical flow field. Specifically, it predicts the parameters for the discretization of Weickert et al. (2013), which ensures that the diffusion evolution is stable and well-posed.

We compare our learned diffusion inpainting network with Edge-Enhancing Diffusion (Weickert, 1994) inpainting, Absolutely Minimizing Lipschitz (Raad et al., 2020) inpainting and Laplace-Beltrami (Raad et al., 2020) inpainting. Additionally, we consider the most popular state-of-the-art deep learning methods that use U-Nets (Ronneberger et al., 2015), Wasserstein GANs (Vašata et al., 2021), and Probabilistic Diffusion (Saharia et al., 2022; Lugmayr et al., 2022) and show that our proposed method can reconstruct flow fields with a high level of detail and generalize exceptionally well. Evaluated with test data from the same domain as the training data, our method achieves an average improvement of $48 - 66\%$ in terms of endpoint error when compared to the baselines, while when tested on a new domain, our method manages to outperform by $11 - 47\%$ and sets a new state of the art. Finally, we evaluate on real world data from autonomous driving and show that in this practical application, our method is on-par with other methods or significantly outperforms them. Beyond the good generalization capabilities, the diffusion networks have comparatively few learnable parameters and competitive inference times. Our ablation studies show that they can be trained with much less data and still outperform baselines trained on the full dataset.

## 1.2 Related Work

**Diffusion inpainting.** Reconstructing missing information from images, known as inpainting (Guillemot & Le Meur, 2014), has been a long-standing goal in image processing (Masnou & Morel, 1998; Bertalmío et al., 2000). For inpainting-based image compression (Galić et al., 2008), diffusion processes offer very good performance. They are theoretically well-founded (Weickert, 1998), and their discretizations are well-understood (Weickert et al., 2013). Moreover, they are inherently explainable and can reconstruct high resolution images in real time (Kämper & Weickert, 2022).

**Inpainting with deep learning.** In recent years, the advances in deep learning methods have shifted the attention towards large-scale data-driven models. Generative Adversarial Networks (GANs) (Vašata et al., 2021) or Probabilistic Diffusion (PD) models (Lugmayr et al., 2022) show impressive inpainting qualities for image restoration and artistic purposes. They do, however, require large amounts of training data and can fail to generalize to out-of-distribution scenarios.

**Inpainting of flow fields.** Inpainting for optical flow fields has rarely been addressed. Jost et al. (2020) investigated PDE-based inpainting for compression of general piecewise smooth vector fields. Andris et al. (2021) use flow field compression and inpainting of optical flow fields as part of their video compression codec. However, both works assume having access to the complete flow to optimize the inpainting mask accordingly, whereas our method works with random, non-optimal masks.

**Relationships between PDE-models and deep learning.** A variety of other neuroexplicit approaches have been explored. Researchers recently have turned to investigating connections between discrete models for solving PDEs and deep learning (Alt et al., 2022; Haber & Ruthotto, 2017; Ruthotto & Haber, 2020; Chen et al., 2018). CNN architectures share a particularly close relationship to discrete PDE models due to the inherent similarity of convolutions and discrete derivatives in the form of finite differences (Morton & Mayers, 2005).

Alt et al. (2022) and Ruthotto & Haber (2020) connected discrete models for solving PDEs and residual blocks (He et al., 2016). Their diffusion blocks realize one explicit step of a discrete diffusion evolution in a residual block with symmetric filter structure. Similar to our approach, they construct architectures that realize diffusion evolutions. However, their work only involves the formulation as a neural network for executing the method (Alt et al., 2022), or the focus revolved around learning the finite differences (Ruthotto & Haber, 2020).

Most closely related to our method are the works of Alt & Weickert (2021) and Chen & Pock (2017) that focused on parameterizing diffusion processes through learning. Both methods use learning to estimate contrast parameters for the diffusivity, but formulate the diffusion tensor as explicit

functions of image contrast. Alt & Weickert (2021) construct a multiscale anisotropic diffusion process for image denoising. Chen & Pock (2017) formulate a general framework for diffusion-reaction systems that support learnable contrast parameters in an isotropic diffusion process and learnable weights for the finite difference operators. However, once trained, these parameters are the same for all pixels and do not adapt themselves to the presented input image content. In contrast, our model learns to drive the explicit diffusion process in a fully neural way and does not rely on first order image derivatives as an edge detector.

## 2 INPAINTING WITH EXPLICIT DIFFUSION

In this section, we review diffusion (Weickert, 1994) and how it can be used for inpainting.

### 2.1 DEFINITION OF DIFFUSION INPAINTING

A given vector valued image $\boldsymbol{f}(\boldsymbol{x}) : \Omega \to \mathbb{R}^c$ is only known on the subset $\Omega_C \subset \Omega$ of the rectangular image domain $\Omega \subset \mathbb{R}^2$. For each channel $i \in \{1, ..., c\}$, diffusion results in the steady state approached for $t \to \infty$ of the initial boundary value problem:

$$\partial_t u_i(\boldsymbol{x}, t) = \mathrm{div}(\boldsymbol{D} \boldsymbol{\nabla} u_i(\boldsymbol{x}, t)) \qquad \text{for} \quad \boldsymbol{x} \in \Omega \setminus \Omega_C \times (0, \infty), \qquad (1)$$

$$u_i(\boldsymbol{x}, t) = f_i(\boldsymbol{x}) \qquad \text{for} \quad \boldsymbol{x} \in \Omega_C \times [0, \infty), \qquad (2)$$

$$u_i(\boldsymbol{x}, 0) = 0 \qquad \text{for} \quad \Omega \setminus \Omega_C, \qquad (3)$$

$$\partial_{\boldsymbol{n}} u_i(\boldsymbol{x}, t) = 0 \qquad \text{for} \quad \boldsymbol{x} \in \partial\Omega \times [0, \infty). \qquad (4)$$

Here, $u_i(\boldsymbol{x}, \infty)$ denotes the final inpainting result in channel $i$, $\mathrm{div} = \boldsymbol{\nabla}^\top$ denotes the spatial divergence operator, and $\partial_{\boldsymbol{n}}$ represents the directional derivative along the normal vector to the image boundary $\partial\Omega$. The diffusion tensor $\boldsymbol{D}$ is a 2×2 positive definite symmetric matrix that describes the propagation behavior.

### 2.2 EDGE-ENHANCING DIFFUSION

Edge-Enhancing Diffusion (EED) (Weickert, 1994) provides superior inpainting quality (Schmaltz et al., 2014), achieved by deriving the diffusion tensor $\boldsymbol{D}$ through the structure tensor (Di Zenzo, 1986):

$$\boldsymbol{S}(u) := \sum_{i=1}^{c} \boldsymbol{\nabla} u_{i,\rho} \boldsymbol{\nabla} u_{i,\rho}^\top. \qquad (5)$$

Here, $u_{i,\rho}$ denotes a convolution of channel $u_i$ with a Gaussian of standard deviation $\rho$. The eigenvalues $\mu_1 \geq \mu_2 \geq 0$ of $\boldsymbol{S}$ measure the local contrast along the corresponding eigenvectors $\boldsymbol{v}_1, \boldsymbol{v}_2$. EED penalises smoothing *across* image structures by transforming the larger eigenvalue with a positive, decreasing diffusivity function $g$. For the remaining direction *along* image structures, full diffusion is allowed by setting the eigenvalue to 1. This results in

$$\boldsymbol{D} := g(\boldsymbol{S}) = g(\mu_1) \cdot \boldsymbol{v}_1 \boldsymbol{v}_1^\top + 1 \cdot \boldsymbol{v}_2 \boldsymbol{v}_2^\top. \qquad (6)$$

In our setting, $\boldsymbol{f}$ is a sparse flow field that should be inpainted. So far, we illustrated a nonlinear diffusion process where the structure tensor from Equation 5 and consequently the diffusion tensor are determined from the evolving signal. This has the disadvantage that the diffusion tensor needs to be re-estimated for every diffusion step. For this reason, we choose a linear diffusion process, and determine the diffusion tensor using the image $\boldsymbol{I}(\boldsymbol{x}) : \Omega \to \mathbb{R}^3$ relative to which the optical flow field is defined. We will refer to $\boldsymbol{I}$ as the reference image.

### 2.3 DISCRETIZATION

Equation 1 can be discretized by means of a finite difference scheme. To transform the continuous into discrete signals, we sample $\boldsymbol{u}, \boldsymbol{f}$ and $\boldsymbol{I}$ at grid sizes $h_x, h_y$. We discretize the temporal derivative by a forward difference with time step size $\tau$. The spatial first-order derivative operator $\boldsymbol{\nabla}$ and its adjoint $\boldsymbol{\nabla}^\top$ are implemented by a convolution matrix $\boldsymbol{K}$ and its negated transpose $-\boldsymbol{K}^\top$, respectively.

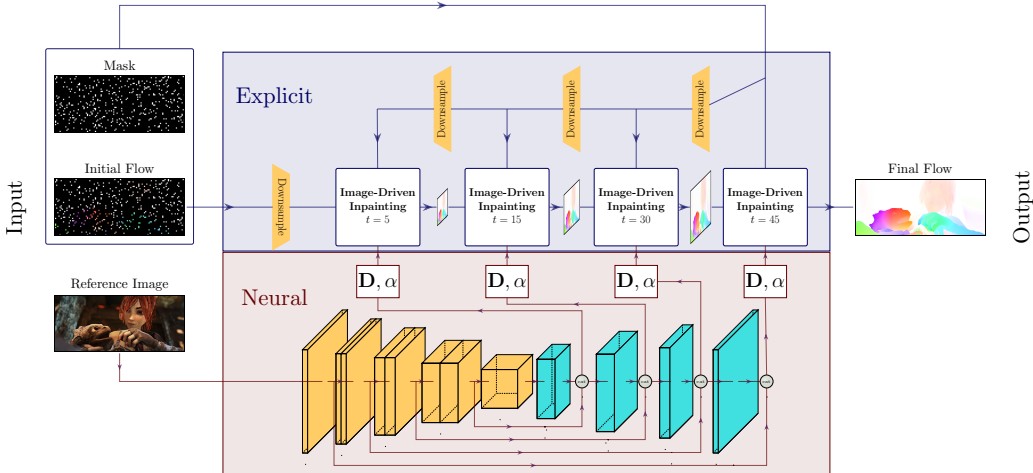

Figure 1: **Our proposed hybrid inpainting model.** The Diffusion Tensor Module takes the reference image as input, and outputs a specific diffusion tensor $\boldsymbol{D}$ and discretization parameter $\alpha$ for every stage of the coarse-to-fine inpainting pipeline. The inpainting itself is done using a stable and well-posed anisotropic diffusion process that solves $t$ steps of the explicit scheme in Equation 8.

To discretize computation and multiplication with the diffusion tensor $\boldsymbol{D}$, we introduce the following notation for an activation function:

$$\Phi(\boldsymbol{I},\ \boldsymbol{K}\boldsymbol{u}^k) = g\Big(\underbrace{\sum_{i=0}^{c}(\boldsymbol{K}\boldsymbol{I})_i(\boldsymbol{K}\boldsymbol{I})_i^{\top}}_{\boldsymbol{S}}\Big)(\boldsymbol{K}\boldsymbol{u}^{\boldsymbol{k}}). \tag{7}$$

Finally, we can define the discrete diffusion evolution and solve it for the next time step to obtain an explicit scheme:

$$\frac{\boldsymbol{u}^{k+1} - \boldsymbol{u}^k}{\tau} = -\boldsymbol{K}^{\top}\Phi(\boldsymbol{I},\ \boldsymbol{K}\boldsymbol{u}^k) \quad \Leftrightarrow \quad \boldsymbol{u}^{k+1} = \boldsymbol{u}^k - \tau(\boldsymbol{K}^{\top}\Phi(\boldsymbol{I},\ \boldsymbol{K}\boldsymbol{u}^k)). \tag{8}$$

where the time levels are indicated by superscripts.

To achieve an inpainting process with good reconstruction qualities, the choice of the convolution matrix $\boldsymbol{K}$ is crucial. Weickert et al. (2013) introduced a nonstandard finite difference discretization that implements the discrete divergence term $\boldsymbol{K}^{\top}\Phi(\boldsymbol{I},\ \boldsymbol{K}\boldsymbol{u}^k)$ on a 3×3 stencil. It introduces two free parameters

$$\alpha \in [0, \tfrac{1}{2}],\ \ |\beta| \leq 1 - 2\alpha, \tag{9}$$

that have an impact on sharpness and rotation invariance of the discretization.

## 3 FROM EXPLICIT TO NEUROEXPLICIT FLOW INPAINTING

In the following paragraphs, we discuss how to transform an explicit diffusion inpainting approach into a hybrid neuroexplicit architecture, where we define explicit as parts of the architecture that are derived from the well known PDE framework for the diffusion process, and neural as generic data-driven non-interpretable deep learning architecture parts.

To solve the inpainting task, our method and the baselines receive a sparse, initial flow field as well as a binary mask that marks the positions of the known flow vectors. Additionally, we provide the reference image for an image-driven inpainting process. Image-driven regularizers in traditional optical flow methods exploit the correlation of contrast in the reference image and the discontinuities in the unknown flow field. However, considering contrast alone is not sufficient and leads to over-segmented flow fields. In practice, due to the aperture problem, one must decide how to regularize

based on the individual image content. As this is of statistical nature and requires prior knowledge, leveraging deep learning here seems adequate. We bring in a U-Net (Ronneberger et al., 2015) as the *Diffusion Tensor Module (DTM)*, that we train end-to-end to predict the ideal diffusion parameters. Concretely, it replaces the heuristic choice of the structure tensor in the activation $\Phi$ in Equation 7 and the discretization parameter $\alpha$ in Equation 9. An overview of the complete model is shown in Figure 1.

### 3.1 COARSE-TO-FINE DIFFUSION INPAINTING

Our architecture implements an explicit image-driven inpainting process. In contrast to traditional methods, the parameters of the process are obtained from the reference image using the DTM. To reduce the required time steps of the inpainting process and make it computationally feasible, we employ a coarse-to-fine scheme (Bornemann & Deuflhard, 1996). We build the pyramid using pooling operations that downsample by a factor 2, where we use average pooling for the reference image and max pooling for the mask. To obtain the coarse version of the sparse flow field, we use average pooling of flow known values. After obtaining the coarse versions of all inputs, we start the diffusion inpainting process from the coarsest sparse flow field. The inpainted flow field is then upsampled using bilinear interpolation and initializes the inpainting process at the next finer resolution.

For a multiscale diffusion process that spans across $N$ resolutions, we need a set of parameters at each scale. The DTM performs $N$ down- and upsampling convolutions. At each feature map after the bottleneck, we apply a separate convolution to estimate a feature map $\boldsymbol{z}$ with five channels. Below, we explain how each $\boldsymbol{z}$ is transformed to parameterize the diffusion process along the coarse-to-fine pyramid.

### 3.2 DISCRETIZATION

To implement our scheme, we use the discretization of Weickert et al. (2013) that we discussed in Section 2.3. This formulation introduces the two free parameters $\alpha$ and $\beta$ shown in Equation 9. The first channel of $\boldsymbol{z}$ is used as the discretization parameter $\alpha = \sigma(z_0)/2$. Notably, the restriction of $|\beta| \leq 1 - 2\alpha$ depends on $\alpha$. To guarantee a stable scheme, we choose $\beta = (1 - 2\alpha)\text{sign}(b)$, where $b$ is the off-diagonal element of the diffusion tensor.

### 3.3 LEARNING THE DIFFUSION TENSOR

The remaining four channels in $\boldsymbol{z}$ are used to estimate the diffusion tensor in the activation $\Phi$. Replacing the structure tensor with a neural edge detector allows learning a prior that decides which image edges will likely coincide with flow discontinuities. Due to the anisotropic diffusion process, these discontinuities can be maintained throughout the inpainting process even if the mask distribution is sparse and suboptimal.

Concretely, we obtain two eigenvalues $\mu_1 = g(z_1)$, $\mu_2 = g(z_2)$ and one eigenvector $\boldsymbol{v}_1 = \frac{(z_3, z_4)^\top}{\|(z_3, z_4)^\top\|_2}$. We explicitly compute $\boldsymbol{v}_2 = \frac{(-z_4, z_3)^\top}{\|(z_3, z_4)^\top\|_2}$ to ensure orthogonality to $\boldsymbol{v}_1$. The eigenvalues are constrained to the range $[0, 1]$ using the Perona-Malik diffusivity $g(x) = (1 + \frac{x^2}{\lambda^2})^{-1}$ (Perona & Malik, 1990), where the free parameter $\lambda$ is learned during training. In contrast to the formulation in Equation 6, we apply the diffusivity to both eigenvalues. This gives the DTM additional freedom to either replicate the behavior in Equation 6 or restrict the diffusive flux in both directions.

## 4 EXPERIMENTS

Our experiments are divided into three parts. In the first part, we compare our method with a selection of fully explicit and neural inpainting methods. In the second part, we show an ablation study to test the effectiveness of learning different components of the diffusion inpainting process to analyze and determine the balance between explicit and neural parameter selection. For both parts, we train on the final subset of the FlyingThings dataset (Mayer et al., 2016). To evaluate generalization, we test on the Sintel dataset (Butler et al., 2012).

Table 1: **Comparison of our method with the baselines on the Sintel dataset.** Surprisingly, the explicit EED diffusion inpainting outperforms the data-driven baselines across both datasets. Learning the proposed parts of the diffusion process with our method further improves the reconstruction qualities with significantly fewer iterations and leads to a new state of the art. *indicates that the method was already tuned for Sintel.

| | Training-Domain Test EPE | | | | | | | Sintel Test EPE | | | | | | |
|---|---|---|---|---|---|---|---|---|---|---|---|---|---|---|
| | EED | AMLE* | LB* | Ours | FlowNetS | WGAIN | PD | EED | AMLE* | LB* | Ours | FlowNetS | WGAIN | PD |
| 1% | 2.06 | 2.03 | 2.03 | 1.01 | 2.33 | 2.26 | 3.96 | 0.94 | 0.94 | 0.86 | 0.72 | 0.85 | 1.14 | 2.39 |
| 5% | 1.00 | 1.08 | 1.00 | 0.55 | 1.68 | 1.73 | 1.09 | 0.52 | 0.51 | 0.43 | 0.40 | 0.57 | 0.80 | 0.55 |
| 10% | 0.73 | 0.82 | 0.75 | 0.39 | 1.55 | 1.43 | 0.72 | 0.43 | 0.38 | 0.31 | 0.28 | 0.51 | 0.60 | 0.40 |

Finally, we demonstrate the usefulness of our method in a real-world application. The KITTI (Geiger et al., 2012) dataset is well known for autonomous driving and provides sparse ground truth that is acquired from registering LiDAR scans. Densifying it presents a practically highly relevant use case of our method.

## 4.1 NETWORK AND TRAINING DETAILS

For our diffusion inpainting network, we leverage four resolutions for the coarse-to-fine pyramid. Going from coarsest to finest, we perform $i$ iterations, where $i \in \{5, 15, 30, 45\}$ increases with the resolution. At each resolution, the feature map $z$ is obtained from the DTM and transformed into the parameters as described in Section 3. Each resolution has access to a separate contrast parameter $\lambda$ in the Perona-Malik diffusivity discussed in Section 3.3, which is initialized as 1 and learned during training. To further speed up the inpainting process, we use the Fast-Semi Iterative (FSI) scheme proposed by Hafner et al. (2016) and perform one cycle per resolution. Time step size and FSI extrapolation weights are chosen to satisfy a stable and well-posed diffusion inpainting process. For more information about the FSI scheme, we refer to the supplementary material.

We choose three fully explicit baselines. First, a linear EED inpainting method where all parameters are chosen explicitly and the diffusion tensor is also estimated using the reference image. For a fair comparison, we use the same coarse-to-fine strategy and optimize its hyperparameters on a subset of the training data and let the inpainting process converge. The previous state of the art is held by Raad et al. (2020), who propsoe two anisotropic optical flow inpainting algorithms: the first is based on the Absolutely Minimizing Lipschitz Extension (AMLE) PDE and the second one uses the Laplace-Beltrami (LB) operator. They propose a set of robust hyperparameters for the Sintel dataset, which we will use for all evaluations. Note that other methods are not trained or tuned on Sintel and therefore this setting gives Raad et al. (2020) an advantage.

As the first neural baseline, we choose a FlowNetS (Dosovitskiy et al., 2015) as a general purpose U-Net architecture. Instead of two images, we feed it a concatenation of image, mask and sparse flow and let the network learn the inpainting process. As more recent and advanced deep learning-based methods, we include Generative Adversarial Networks (GANs) and Probabilistic Diffusion (PD). We use WGAIN (Vašata et al., 2021) as the GAN baseline, as it has been used successfully employed for inpainting images from sparse masks. For the PD network, we adapted the popular efficient U-Net architecture (Saharia et al., 2022) and use the inpainting formulation of RePaint (Lugmayr et al., 2022) during inference. Both GAN and PD network are conditioned on the reference image to learn the correlation between flow and image edges. For more details on training and adaptations, we refer to the supplemental.

## 4.2 RECONSTRUCTION AND GENERALIZATION

Table 1 shows a comparison of our method to the introduced baselines. Edge-Enhancing Diffusion (EED) inpainting outperforms the purely neural baselines and can reconstruct a high level of detail. It does, however, require a significant number of iterations to converge (anywhere from 3,000 to 100,000), varying drastically with the content of the images and the given mask density. The reason for this slow convergence is the reliance on the structure tensor that we discussed in Section 2.2. The diffusivity is limiting the diffusive flux wherever there is image contrast, which increases the number of required time steps. Figure 4 shows that relying on the structure tensor can also be

harmful in low-contrast regions where no edge is identified and information leaks across edges. Since we let the inpainting process fully converge, this leaking effect can be detrimental to the final performance. Furthermore, replacing the structure tensor with a neural edge detector leads to a more robust inpainting in cases where there is a high variance of contrast within the images, which happens frequently in the FlyingThings data. Consequently, the discrepancy between EED and our method is much more severe on that dataset.

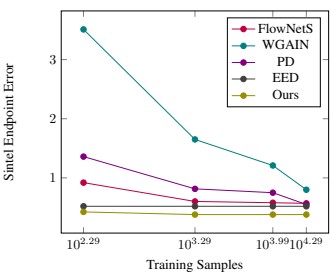 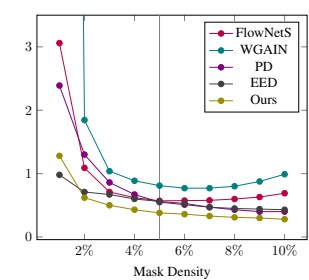 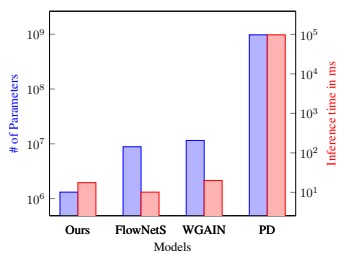

Figure 2: **Our proposed method is robust to changes in the training and inference setting.** The left plot shows the weaker reliance of our method on training data. Using $194$ samples, we reach a competitive performance to the network trained on the full dataset. The right plot shows the favorable generalization to unseen mask densities of our method and the explicit EED inpainting. We evaluated each model optimized for $5\%$ on previously unseen mask densities.

Figure 3: **Weights and inference time of the models.** Compared to the baselines, our model is very lightweight and has competitive inference times. Notably, we omitted the explicit baselines, since there is no clear way to compare the methods.

As can be seen in Figure 4, the CNN methods FlowNetS (Dosovitskiy et al., 2015) and WGAIN (Vašata et al., 2021) produce noisy flow fields at the flow edges. They fail to capture the same level of detail as the anisotropic diffusion methods that adapt their smoothing behavior to the image content. Compared to the CNN methods, the Probabilistic Diffusion (PD) model with the RePaint (Lugmayr et al., 2022) inpainting has well-localized discontinuities. However, PD models are highly affected by the distribution of the training data. Figure 4 shows a case of overfitting in the first two rows. The FlyingThings dataset contains mostly rigid objects with straight edges and no materials comparable to the fuzzy beard of the shaman. Consequently, PD fails to generalize to the out-of-domain sample and reconstructs a blocky and unnatural looking beard.

Since our method realizes a well-posed diffusion process by construction, it is naturally robust to changes in its input. We tested the generalization capabilities to new mask densities and show the results in Figure 2. Increasing the mask density compared to observed training density should lead to an increase in performance since more information is presented. The data-driven baselines that are trained with a specific density (FlowNetS and WGAIN) fail to capture this intuition and have decreasing performance. Our proposed method has an increasingly better reconstruction quality with higher densities. When evaluated on a density of $10\%$, the network trained on $5\%$ density can even reach a very close EPE on to the network that was optimized on this density ($0.28$ vs. $0.29$).

Figure 3 shows the number of learned parameters of all models. Since the inpainting behavior in our method is steered by an explicit anisotropic diffusion process, the network has significantly fewer parameters than the compared baselines. A vast majority of these parameters are placed in the DTM to identify flow discontinuities and drive the diffusion process accordingly. Having so few parameters provides an inherent regularizing effect and leads to less reliance on available training data. This is reflected in the left plot in Figure 2, where we compared the performance of all baselines when trained on a subset of the available training data. Even with a drastic cut of training data, our proposed method outperforms all other baselines. The reconstruction quality barely decreases compared to the networks that were trained on more data.

### 4.3 EFFECTS OF LEARNED COMPONENTS

Table 2 shows quantitative results of trained inpainting networks, where we illustrate the effect of learning the eigenvalues $\mu_1, \mu_2$, eigenvectors $\boldsymbol{v}_1, \boldsymbol{v}_2$, discretization parameter $\alpha$. As one can see from Equation 5, the structure tensor computation considers only first-order image derivatives. Especially

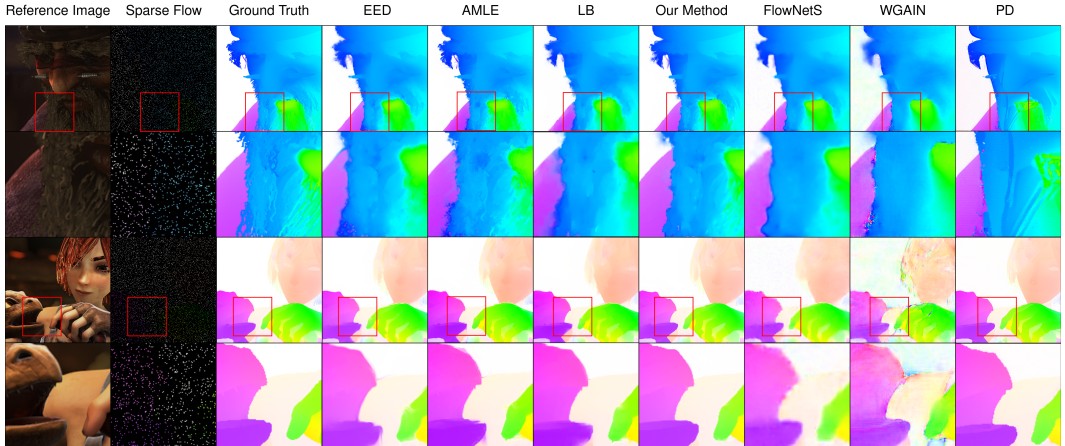

Reference Image   Sparse Flow   Ground Truth   EED   AMLE   LB   Our Method   FlowNetS   WGAIN   PD

Figure 4: **Samples generated with a mask density of** $5\%$**.** Every other row displays the zoomed in area of the red rectangle in the row above. Our method manages to retain a much higher level of detail in the reconstructed flow fields. In the bottom row at the dragons chin, we can observe that the PDE methods (EED, AMLE, and LB) fail to maintain flow edges in low contrast regions. Notably, both WGAIN and the PD model have poor out-of-distribution performance. WGAIN tends to have large outliers and fails in the zero flow in the background. The PD model fails to reproduce the fuzzy material of the shamans beard due to a lack of comparable materials in the training data.

Table 2: **Ablation study of replacing learned with explicit components.** The values indicate endpoint errors relative to our final network for different mask densities. Learning the eigenvalues over explicit ones $(-\mu)$ plays the biggest role. Exchanging learned with explicit eigenvectors $(-\boldsymbol{v})$ and the learned, spatially-varying discretization parameter with a constant explicit one $(-\alpha)$ leads to consistent decreases in performance. Learning the second discretization parameter $(+\beta)$ can further push the performance, but does not guarantee stability. We also test the ResNet implementation of Alt et al. (2022) and learn the finite difference operators $(+\boldsymbol{W})$. However, this does not give a significant performance improvement and leads to additional computation cost.

|  | Training-Domain Test EPE | | | | | | Sintel Test EPE | | | | |
|---|---|---|---|---|---|---|---|---|---|---|---|
|  | Full | $-\mu$ | $-\boldsymbol{v}$ | $-\alpha$ | $+\beta$ | $+\boldsymbol{W}$ | Full | $-\mu$ | $-\boldsymbol{v}$ | $-\alpha$ | $+\beta$ | $+\boldsymbol{W}$ |
| 1% | 1.01 | +0.95 | +0.07 | +0.05 | −0.04 | +0.00 | 0.72 | +0.17 | +0.07 | +0.03 | −0.02 | +0.01 |
| 5% | 0.55 | +0.28 | +0.02 | +0.06 | −0.03 | −0.01 | 0.40 | +0.04 | +0.06 | +0.02 | −0.01 | −0.01 |
| 10% | 0.39 | +0.20 | +0.00 | +0.02 | −0.01 | +0.03 | 0.28 | +0.04 | +0.04 | +0.02 | −0.01 | +0.00 |

in the presence of noisy images, the structure tensor relies on Gaussian pre-smoothing that can lead to worse edge localization in the final flow field. In the case of explicit eigenvalues, the network has to learn a small contrast parameter $\lambda$ to avoid smoothing across structures. This leads to a slow convergence within the limited number of performed diffusion steps and poor inpainting quality. Supplying the eigenvectors of the diffusion tensor by the learnable module provides a consistent increase in performance.

The discretization parameters are usually chosen as constant hyperparameters, whereas our DTM outputs one $\alpha$-value per pixel and outperforms the global constant for all mask densities. This suggests that adapting the discretization parameters to the image content is preferable to obtain high-quality reconstructions. We also performed an additional ablation study to estimate the second discretization parameter $\beta$ independently of $\alpha$. The improvements are insignificant, while doing so voids the restriction 9. Hence, we do not recommend learning the $\beta$.

Alt et al. (2022) showed that for each discrete diffusion evolution with a fixed stopping time, there is an equivalent ResNet architecture that implements it. We additionally compare our method to the ResNet formulation of Alt et al. (2022) where we learned the discrete derivative operators. This formulation requires much more arithmetic operations compared to the efficient 3×3-stencil of We-ickert et al. (2013), since each derivative operator has to be realized with its own convolution kernel. On the other hand, also in this case, the performance difference is insignificant and the explicit dis-

Table 3: **Generalization Capabilities to Real World Data.** We report the combined EED on all measured pixels in the original KITTI training dataset. As customary for KITTI, we additionally report flow (FL) outliers in % as defined by Geiger et al. (2012). A displacement is considered an outlier if its endpoint error is $> 3$, or it differs by at least $5\%$ of the ground truth displacement. The results show that our method is on par with Laplace-Beltrami in terms of EPE but has significantly fewer outliers especially in the most challenging low density setting.

|  | EPE | | | | | | FL | | | | | |
|---|---|---|---|---|---|---|---|---|---|---|---|---|
|  | EED | AMLE | LB | Ours | FlowNetS | WGAIN | EED | AMLE | LB | Ours | FlowNetS | WGAIN |
| 1% | 1.11 | 1.26 | 1.07 | 1.07 | 1.43 | 3.18 | 1.14 | 1.19 | 0.94 | 0.87 | 1.2 | 3.59 |
| 5% | 0.46 | 0.56 | 0.46 | 0.47 | 2.53 | 7.0 | 0.27 | 0.35 | 0.26 | 0.25 | 2.26 | 4.48 |
| 10% | 0.23 | 0.30 | 0.23 | 0.23 | 4.96 | 6.82 | 0.11 | 0.16 | 0.11 | 0.11 | 3.77 | 4.39 |

cretization from Weickert et al. (2013) is preferable. For the derivation of the equivalent ResNet architecture, we refer the reader to the supplementary material.

## 4.4 GENERALIZATION TO REAL WORLD DATA

We further tested the generalization capabilities of all methods to real world data using the KITTI2015 (Geiger et al., 2012) optical flow dataset. This dataset provides accurate sparse measurements obtained from registering LiDAR scans. Densifying these measurements resembles a highly relevant application of our method for autonomous driving.

The original density of the measurements is between $15 - 25\%$. To measure the accuracy of our reconstructed dense flow fields, we subsample according to our different density settings $1$, $5$ and $10\%$. After reconstructing the dense flow field, we then measure the accuracy of the previously left out measurements and report the results in Table 3. Please note that we omit the Probabilistic Diffusion method, since it is optimized for a specific image resolution and fails to generalize from the square training images to the wide-angle setting in KITTI.

When looking at the numbers, the advantage in terms of robustness of all the PDE-based, neural and neuroexplicit methods becomes apparent. The neural models fail to generalize to this new real-world setting, as well as non-uniform mask distributions. This is consistent with the observations from Figure 2. The results show that our method which combines neural and explicit components is on-par with Laplace-Beltrami in terms of EPE, but has significantly fewer outliers especially in the most challenging low density setting. Most likely, our method could still be improved by supplying different non-uniform mask distributions during training to adapt to the setting in KITTI.

## 5 CONCLUSION AND FUTURE WORK

We studied discrete diffusion processes in a deep learning context and illustrated a novel approach to steer the diffusion process by a deep network. We showed that our method can outperform both model-based and fully data-driven baselines, while requiring less training data, having fewer parameters, generalizing better, being well-posed, being supported by a stability guarantee, and offering competitive runtimes. Our work shows that combining general learning-based methods with specialized mathematical models can lead to performant hybrid networks that inherit the best of both worlds, and motivates further research on neuroexplicit models.

In the current work, we only focus on the regularization aspect of optical flow. Future work will be on embedding our diffusion regularization into an end-to-end optical flow algorithm. We expect that this will yield more interpretable methods with the benefits of stability guarantees, better generalization and requiring less training data.

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

## A    TRAINING DETAILS

In this section, we provide further details about the training procedure of our method. Unless otherwise specified, these details also apply to the considered baselines we will discuss in later sections.

We evaluate our methods on a combination of two datasets. We train on the FlyingThings3D (Mayer et al., 2016) subset that removes overly large displacements and evaluate on the Sintel dataset (Butler et al., 2012). For both datasets, we use the final versions that include image degradations such as motion or depth-of-field blur. As the evaluation metric, we choose the average Endpoint Error (EPE), as it is customary for optical flow evaluation. While training, the mask is drawn from a uniform distribution and binarized to adhere to the desired density. During evaluation, we use a fixed mask for each image. To speed up the training, we use center-cropped images of size 384×384 pixels.

All methods are implemented in PyTorch and trained on an Nvidia A100 GPU. We trained the neural networks for a total of 900,000 iterations using a batch size of 16. For the optimizer, we choose Adam (Kingma & Ba, 2015) with the default parameter configuration $\beta_1 = 0.9, \beta_2 = 0.999$. We use an initial learning rate of 0.0001 that is halved every 100,000 iterations after the first 300,000.

## B    IMPLEMENTATION DETAILS

In this section, we provide some additional implementation details of our methods.

The CNN architecture we use to estimate parameters to the diffusion process is a simple UNet (Ronneberger et al., 2015) architecture. Table 4 shows the layers and corresponding channel dimensions in our Diffusion Tensor Module.

### B.1    FAST-SEMI ITERATIVE SCHEME

In practice, enforcing the stability of an explicit scheme requires restricting the time step $\tau$. However, depending on the data, a stable scheme may require a substantial amount of iterations to converge. Formulated as a neural architecture, this results in an excessive amount of layers and intractable optimization. Fortunately, explicit schemes can be accelerated by extrapolating the outcome of each time step, e.g. by the Fast Semi-iterative (FSI) Scheme proposed by Hafner et al. (2016). In contrast to a naive discretization, FSI-schemes implement a cycle of $L$ steps and extrapolate the diffusion result at a fractional time step $k + \frac{l-1}{L}$:

$$\boldsymbol{u}^{k+\frac{l+1}{L}} = \gamma_l(\boldsymbol{u}^{k+\frac{l}{L}} + \tau(-\boldsymbol{K}^\top(\Phi(\boldsymbol{I}, \boldsymbol{K}\boldsymbol{u}^{k+\frac{l}{L}}))) + (1 - \gamma_l)\boldsymbol{u}^{k+\frac{l-1}{L}} \tag{10}$$

with $l = 0, ..., L - 1$ indexing the step and $\gamma_l := (4l + 2)/(2l + 3)$ denoting the time-varying extrapolation weights.

### B.2    RESNET IMPLEMENTATION OF DIFFUSION EVOLUTIONS

As stated by Alt et al. (2022), a discretization of any higher order diffusion evolution can be formulated as a variant of a ResNet (He et al., 2016) block. Alt et al. (2022) introduced their diffusion blocks for nonlinear diffusion processes where the diffusion tensor is determined from the evolving signal. To translate ResNet into diffusion blocks, we will follow their formulation and consider such a diffusion process. However, in the following subsection we will show how to adapt these diffusion blocks into the linear, image-driven diffusion process we considered as part of our ablation study.

Starting from the conventional version of a ResNet block, we are now constructing a nonlinear diffusion process from it. To this end, we reintroduce the activation function we used in the main section of the paper:

$$\Phi(\boldsymbol{K}\boldsymbol{u}) = g\Big(\underbrace{\sum_{i=0}^{c}(\boldsymbol{K}\boldsymbol{u})_i(\boldsymbol{K}\boldsymbol{u})_i^\top}_{S}\Big)(\boldsymbol{K}\boldsymbol{u}). \tag{11}$$

Note the change in notation, where we removed the first argument that determines the structure tensor. We do this, to make the translation into the ResNet block more intuitive.

Table 4: Architecture of the Diffusion Tensor Module

| Encoder | | | | Decoder | | | |
|---|---|---|---|---|---|---|---|
| Layer | in-ch | out-ch | stride | Layer | in-ch | out-ch | input |
| cnv0 | 3 | 44 | 1 | dcnv4 | 352 | 176 | cnv4 |
| cnv1 | 44 | 44 | 2 | dcnv3 | 352 | 176 | dcnv4+cnv3_1 |
| cnv1_1 | 44 | 44 | 1 | dcnv2 | 264 | 88 | dcnv3+cnv2_1 |
| cnv2 | 44 | 88 | 2 | dcnv1 | 132 | 44 | dcnv2+cnv1_1 |
| cnv2_1 | 88 | 88 | 1 | dt4 | 352 | 5 | dcnv4+cnv3_1 |
| cnv3 | 88 | 176 | 2 | dt3 | 352 | 5 | dcnv3+cnv2_1 |
| cnv3_1 | 176 | 176 | 1 | dt2 | 264 | 5 | dcnv2+cnv1_1 |
| cnv4 | 176 | 352 | 2 | dt1 | 132 | 5 | dcnv1+cnv0 |

A normal ResNet block can be written in the following form:

$$\boldsymbol{u} = \sigma_2(\boldsymbol{f} + \boldsymbol{W}_2\sigma_1(\boldsymbol{W}_1\boldsymbol{f} + \boldsymbol{b}_1, \, \boldsymbol{y}) + \boldsymbol{b}_2, \, \boldsymbol{y}), \tag{12}$$

with $\boldsymbol{W}_1, \boldsymbol{W}_2$ denoting the application of a convolution kernel, $\boldsymbol{b}_1, \boldsymbol{b}_2$ denoting the respective biases, and $\sigma_1, \sigma_2$ denoting arbitrary activation functions, such as the ReLU (Nair & Hinton, 2010).

$$\sigma_1(\boldsymbol{x}) = \tau\Phi(\boldsymbol{x}), \qquad \sigma_2(\boldsymbol{x}) = \boldsymbol{x}, \qquad \boldsymbol{W}_1 = \boldsymbol{K}, \qquad \boldsymbol{W}_2 = -\boldsymbol{K}^T \tag{13}$$

and with $\boldsymbol{b}_1 = \boldsymbol{b}_2 = \boldsymbol{0}$, we can transform our diffusion process into a ResNet architecture.

Notably, the convolution kernels share their weights since they implement the same operator $\boldsymbol{K}$. In practice, this is implemented by maintaining one kernel $\boldsymbol{W}$ that resembles the inner convolution. The outer convolution kernel can be obtained by mirroring and negating $\boldsymbol{W}$ (Alt et al., 2022). When dealing with more than one input channel, the PDE formulation suggests that inter-channel communication should only happen through the joint diffusion tensor in the activation and not the derivative (e.g. the convolution). This behavior can be realized by implementing the convolution kernel $\boldsymbol{W}$ as grouped convolutions (Krizhevsky et al., 2012) with an equal number of groups to channels.

When learning the finite difference operators in the diffusion block, stability of the diffusion process can be harmed. To avoid this, Alt et al. (2022) suggested a weight normalization process that rescales the convolution kernels after each optimization step. The stability assumptions hold, as long as the maximal absolute eigenvalue of $\boldsymbol{K}$ is less or equal to $1$. In practice, this constraint can be satisfied by rescaling each grouped convolution by $\sqrt{C}\|\boldsymbol{W}\|_2^2$, where $C$ is the number of channels of the considered signal $\boldsymbol{u}$. For more information about diffusion blocks, we kindly refer the reader to the original publication (Alt et al., 2022).

Table 5: All required convolution operators per diffusion block. The outer convolution is obtained by mirroring around the center of the kernel and multiplying by $-1$.

$$\boldsymbol{W}_x^1 = \begin{array}{|c|c|} \hline -1 & 1 \\ \hline 0 & 0 \\ \hline \end{array} \qquad \boldsymbol{W}_x^2 = \begin{array}{|c|c|} \hline 0 & 0 \\ \hline -1 & 1 \\ \hline \end{array} \qquad \boldsymbol{W}_y^1 = \begin{array}{|c|c|} \hline -1 & 0 \\ \hline 1 & 0 \\ \hline \end{array} \qquad \boldsymbol{W}_y^2 = \begin{array}{|c|c|} \hline 0 & -1 \\ \hline 0 & 1 \\ \hline \end{array}$$

### B.3 DIFFUSION BLOCK FORMULATION OF OUR SCHEME

To translate our considered diffusion process into a ResNet-style architecture, we need to construct the diffusion blocks that correspond to the discretization of Weickert et al. (2013) and adapt the activation function shown in Equation 11.

Adapting the activation is straightforward. We only need to go back to our initial formulation in the main section of the paper and let the diffusion block accept an additional input, which corresponds to the structure tensor of the reference image. Since the reference image does not change during the diffusion evolution, the diffusion tensor can be precomputed for each level in the coarse-to-fine

pyramid, allowing for a more efficient scheme. By accepting the reference image in the activation, it brings us back to the original definition of the activation function $\Phi(\boldsymbol{I},\ \boldsymbol{K}\boldsymbol{u})$.

To design the diffusion blocks, we need to decompose the 3×3-stencil of Weickert et al. (2013) into the individual finite difference operators. Weickert et al. (2013) propose a weighted average of two 2×2 for each $x-$ and $y-$derivative. Consequently, they leverage 4 total finite differences per discrete gradient operator. Therefore, each diffusion block requires 4 convolution kernels. The required kernels are shown in Table 5.

Let in the following $\boldsymbol{D} = \begin{pmatrix} a & b \\ b & c \end{pmatrix}$ denote the considered diffusion tensor based on the reference image $\boldsymbol{I}$. The discrete divergence term is then implemented as

$$\boldsymbol{K}^{\top}\Phi(\boldsymbol{I},\ \boldsymbol{K}\boldsymbol{u}) = \boldsymbol{w}^{\top}\boldsymbol{H}\boldsymbol{w}, \tag{14}$$

where $\boldsymbol{w} := (\boldsymbol{W}_x^1\boldsymbol{u}, \boldsymbol{W}_x^2\boldsymbol{u}, \boldsymbol{W}_y^1\boldsymbol{u}, \boldsymbol{W}_y^2\boldsymbol{u})^{\top}$. The construction of the matrix $\boldsymbol{H}$ introduces the discretization parameters $\alpha$ and $\beta$ which will be used to weight the influence of the finite difference operators:

$$\boldsymbol{H} := \begin{pmatrix} \frac{1-\alpha}{2}a & \frac{\alpha}{2}a & \frac{1-\beta}{4}b & \frac{1+\beta}{4}b \\ \frac{\alpha}{2}a & \frac{1-\alpha}{2}a & \frac{1+\beta}{4}b & \frac{1-\beta}{4}b \\ \frac{1-\beta}{4}b & \frac{1+\beta}{4}b & \frac{1-\alpha}{2}c & \frac{\alpha}{2}c \\ \frac{1+\beta}{4}b & \frac{1-\beta}{4}b & \frac{\alpha}{2}c & \frac{1-\alpha}{2}c \end{pmatrix} \tag{15}$$

In this formulation, the role of the discretization parameters also become more clear. $\alpha$ and $\beta$ are used to determine the relative importance of the diagonal and off-diagonal entries of the diffusion tensor respectively. For more details about the discretization, we refer the reader to the original publication of Weickert et al. (2013).

Although this scheme can be implemented very efficiently in its traditional form, translating it into a diffusion block introduces a severe computational overhead. When considering the original stencil, one diffusion step requires a single 3×3 convolution of the input signal. In the diffusion block formulation where each finite difference operator can be learned, one diffusion step requires a total of 8 2×2 convolutions. This does not only slow down the effective inference time, it also bloats the computational graph severely when optimizing each convolution kernel. Since we also did not see a meaningful performance benefit to learning the operators, we opted against the use of the diffusion blocks in our final model.

## C  BASELINE DETAILS

### C.1  EDGE-ENHANCING DIFFUSION INPAINTING

We implement the EED inpainting baseline in PyTorch using the same discretization of Weickert et al. (2013). The reference images are normalized to the range $[0, 1]$. For faster convergence, we use the Fast-Semi-Iterative (FSI) scheme (Hafner et al., 2016) and the same coarse-to-fine setup as in our method. However, instead of a fixed number of iterations as in our learned approach, we let each the inpainting process converge at each resolution. We determine a sufficiently converged state by observing the relative residual and stop the inpainting once it has decreased below $10^{-6}$.

To make for a fair comparison with the deep learning methods, we optimize the free parameters $\lambda, \alpha$ for each considered density on a subset of the training data and keep them fixed during evaluation. Parameters are determined via grid search on 128 samples and we consider the best parameters as the ones that minimize the EPE with the ground truth. Compared to the final evaluation, we stop the inpainting during the parameter optimization once the relative residual decreased by $10^{-5}$. We show the chosen parameter per resolution and the considered interval in Table 6

### C.2  FLOWNETS TRAINING DETAILS

In the inpainting setting we start with a sparse initialization of correct displacements, whereas FlowNetS (Dosovitskiy et al., 2015) needs to find identifiable correspondences given sequential images. Consequently, with over 15.6 million parameters FlowNetS might be unneccessarily complex for our inpainting task. In its original form, FlowNetS estimates the flow at a lower spatial

Table 6: Grid Search to determine the optimal EED-parameters for each density. Step denotes how many evenly spaced values we consider within the search space.

| Parameter | 10% | 5% | 1% | Search Space | Step |
|-----------|-----|-----|-----|--------------|------|
| $\lambda$ | $10^{-4}$ | $10^{-4}$ | $10^{-4}$ | $[10^{-6}, 10^{-2}]$ | 9 |
| $\alpha$ | 0.1 | 0.3 | 0.42 | $[0.001, 0.5]$ | 14 |

resolution and upsamples the initial estimation with bilinear interpolation. This was done to achieve optical flow estimation in real time. Since runtime is not a critical factor for us, we extend the decoder to the full output resolution with two additional transposed convolution layers. The number of channels per layer is reduced throughout the whole network, such that we end up with roughly 8.8 million learnable parameters.

To train the network, we follow mostly the same approach as discussed in A. In addition to that, we added weight decay with weighting parameter 0.0004 and a deep supervision approach for the loss as proposed in (Dosovitskiy et al., 2015). Concretely, this means that we predict a (low resolution) flow at the last 4 layers in the decoder and compute the EPE with downsampled versions of the flow. All losses are aggregated as a weighted combination, where we used the weights $[0.32, 0.08, 0.04, 0.02, 0.01]$ going from coarse to fine resolution.

### C.3 WGAIN TRAINING DETAILS

WGAIN (Vašata et al., 2021) does not adapt well to the flow setting in its original form. We noticed extremely unstable training with diverging loss after a few hundred iterations. We suspect, that this is due to the combination of dealing with (potentially large) flow values and the gradient clipping introduced in (Arjovsky et al., 2017) and used in (Vašata et al., 2021). As a way of mitigating the outliers in the flow, we divide the flow fields by 100 to largely contain them in the range of $[-1, 1]$, but keep the relative distribution the same. To stabilize the training, we replaced the gradient clipping operation with a gradient penalty term (Gulrajani et al., 2017) in the training objective. As discussed in (Gulrajani et al., 2017), we also introduce layer normalization (Ba et al., 2016) in the critic for additional stability. The generator remained largely unchanged, with the exception of the removal of the hard-sigmoid function after the output layer. We adopted the rest of the training procedure from (Vašata et al., 2021), with the exception of using the EPE instead of the Mean Absolute Error (MAE) and choosing $\lambda_g = 1$. The model was trained for the same number of iterations as our method.

### C.4 PROBABILISTIC DIFFUSION TRAINING DETAILS

With the exception of RePaint (Lugmayr et al., 2022), sparse mask inpainting with probabilistic diffusion has rarely been addressed. Since RePaint is only applied during inference, any type of PD model for conditional image generation can be used for our task. We chose the efficient UNet architecture of Imagen (Saharia et al., 2022) and adopted their cascading image generation pipeline. They propose to generate a low-resolution image initially and compose super-resolution models to transform it to the desired resolution. In our case, we generate the initial image at resolution 96×96 and chain one super-resolution net to obtain the final flow at resolution 384×384.

Both networks obtain the reference image as conditioning signal and are otherwise trained for conditional image generation. We used the proposed training parameters in (Saharia et al., 2022), but observed suboptimal results and slow convergence during inference times. Consequently, we adopted the novel training procedure from (Karras et al., 2022) which yielded more effective training and significantly reduced the sampling time during inference. As can be seen in Table 7, this work adds several parameters to control the noise distribution. We kept most of them the same as the optimal parameters in (Karras et al., 2022), but we noticed some improvements by increasing $\sigma_{max}$ and $\sigma_{data}$.

During inference, we perform 48 sampling steps and apply the RePaint (Lugmayr et al., 2022) inpainting at both resolutions. RePaint introduces two parameters, the number of resampling steps and the jump length. In (Lugmayr et al., 2022) the jump length was introduced to avoid blurred

Table 7: Added hyperparameters for training and inference of our Probabilistic Diffusion baseline

| Parameter | Value | Source |
|-----------|-------|--------|
| $\sigma_{min}$ | 0.002 | Karras et al. (2022) |
| $\sigma_{max}$ | $(120, 480)$ | Karras et al. (2022) |
| $\sigma_{data}$ | 1 | Karras et al. (2022) |
| $\rho$ | 7 | Karras et al. (2022) |
| $P_{mean}$ | $-1.2$ | Karras et al. (2022) |
| $P_{std}$ | 1.2 | Karras et al. (2022) |
| $S_{churn}$ | 80 | Karras et al. (2022) |
| $S_{tmin}$ | 0.05 | Karras et al. (2022) |
| $S_{tmax}$ | 50 | Karras et al. (2022) |
| $S_{noise}$ | 1.003 | Karras et al. (2022) |
| Jump Length | 1 | Lugmayr et al. (2022) |
| Resampling Steps | 45 | Lugmayr et al. (2022) |

outputs. However, we observed sharp edges with a jump length of 1 and therefore kept this parameter fixed. The resampling steps, on the other hand, are more critical. They provide a tradeoff between added runtime during sampling and increased conditioning on the known pixels. In (Lugmayr et al., 2022) the masks were dense compared to our setting. We noticed that the proposed number of 10 resampling steps in (Lugmayr et al., 2022) yields poor inpainting quality with mask densities below 10%. To achieve competetive performance on low densities, we had to increase the number of steps and lower the inference time even further. We show the additional parameters we used in Table 7.

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
