# OpenReview forum: "Neuroexplicit Diffusion Models for Inpainting of Optical Flow Fields"
_ICLR.cc/2024/Conference — ICLR 2024 Conference Withdrawn Submission_

### Official Review · Reviewer_cyeS · 2023-10-31

**Soundness:** 2 fair
**Presentation:** 2 fair
**Contribution:** 2 fair
**Rating:** 3
**Confidence:** 3

**Summary:**

The paper proposes a neuroexplicit diffusion model for the optical flow inpainting task. The method combines domain knowledge (explicit PDE-based formulation) with CNN for the task and demonstrates outperforming other baselines such as CNN-based, GAN-based, and probabilistic diffusion baselines.

**Strengths:**

- Good clarity

  The paper includes sufficient details for understanding the main methods (equations, network architecture details, and implementation details). This helps the reproduction of the method.

- Better accuracy over baselines

  The paper compares its method with several baselines (FlowNetS, WGAIN, EED, and PD) and achieves better accuracy than them.

**Weaknesses:**

- Limited evaluation

  The paper evaluates the method only on one synthetic dataset, Sintel. To ensure the method also works on real-world domains, it would be great to evaluate the method on other datasets such as KITTI, Middlebury, etc. Furthermore, the paper doesn't compare with any previous optical flow inpainting methods (eg., Raad et al, "On Anisotropic Optical Flow Inpainting Algorithms"). Achieving better accuracy than baselines is great, but a comparison with previous work would be also necessary to see where the methods stand among the previous works.

  One could also adopt baselines from the depth completion tasks (https://www.cvlibs.net/datasets/kitti/eval_depth.php?benchmark=depth_completion), train their models on the optical flow tasks, and compare with them.

  The method sounds okay, but due to the limited evaluation, it's difficult to judge if the method is really valuable for the community.


- Other applications

  Despite that the proposed method could be generic, the paper demonstrates only optical flow inpainting as an application. Can this method also be applied to other tasks such as depth completion or semantic scene completion? If the paper showed its applicability to such tasks, it could have demonstrated better impact.

**Questions:**

- Transparency?

  What's the meaning of the transparency of the model in the abstract?

---

> ### Author Response · Authors · 2023-11-23
>
> We sincerely thank the reviewer for the effort and appreciate the detailed, valuable feedback.
>
> **W1:** *"The paper evaluates the method only on one synthetic dataset, Sintel. To ensure the method also works on real-world domains, it would be great to evaluate the method on other datasets such as KITTI"*
>
> We agree that more results to demonstrate the effectiveness of our approach in a practical application strengthen the paper. The ground-truth optical flow in the KITTI dataset is acquired from the registration of LiDAR scans, and therefore inherently sparse in its nature. Densifying it presents a practically highly relevant use case of our method. We have evaluated all our methods on this task, including the previous state-of-the-art of inpainting optical flow from random masks. We provide the results in the new Section 4.4 and Table 3 in the paper. Our method is on par with the Laplace-Beltrami method in terms of EPE but significantly outperforms most other methods in the FL metric - especially in the most difficult 1% density case - indicating that our results have fewer outliers. It demonstrates the impact of our method on this selected practical application. Please note that our method was not even adapted for the spatially varying mask densities present in this setting.
>
>
> **W2:** *"Furthermore, the paper doesn't compare with any previous optical flow inpainting methods (eg., Raad et al [1], "On Anisotropic Optical Flow Inpainting Algorithms")"*
>
> We thank the reviewer for directing our attention to this work. We agree that the mentioned method is indeed the prior state of the art. We have evaluated it and show that we significantly outperform it, thus setting a new state of the art. Please see the section “Comparison against the state-of-the-art baseline” in the general comment to all reviewers.
>
> **W3/W4:** *"One could also adopt baselines from the depth completion tasks (https://www.cvlibs.net/datasets/kitti/eval_depth.php?benchmark=depth_completion), train their models on the optical flow tasks, and compare with them. / Despite that the proposed method could be generic, the paper demonstrates only optical flow inpainting as an application. Can this method also be applied to other tasks such as depth completion or semantic scene completion? If the paper showed its applicability to such tasks, it could have demonstrated better impact"*
>
> We agree to the reviewer that this is another application domain for our method. So far, we have provided a methodically novel approach and demonstrated state-of-the-art results for the optical flow inpainting task as well as a positive impact on a practical real-world application. We consider adaptation and evaluation of our approach on depth maps as interesting future work, e.g. as an addition for a journal paper. We thank the reviewer for this feedback and plan to carry out evaluations for the suggested additional application in the near future.
>
> **Q1:** *"What's the meaning of the transparency of the model in the abstract?"*
>
> With transparency, we refer to the fact that the inpainting process is purely done by a diffusion process. Restricting the parameters of the diffusion process to theoretical bounds nets stability guarantees for the inpainting layers. Consequently, the combined model inherits all the mathematical foundations of discrete diffusion processes by construction.
> Therefore, it is straightforward to reason about the behavior of the method a priori.
> It is possible to extract a valid flow field at each step of the inpainting process and judge its quality, due to the fact that we never move the input into a higher dimensional latent space as it would be done with neural inpainting methods.
> This explainable behavior is desirable over black box models for high risk tasks such as autonomous driving.
>
>
> **References:**
>
> [1] Lara Raad, Maria Oliver, Coloma Ballester, Gloria Haro, and Enric Meinhardt. On anisotropic optical flow inpainting algorithms. Image Processing On Line, 10:78–104, 2020

---

### Official Review · Reviewer_3rtj · 2023-10-31

**Soundness:** 3 good
**Presentation:** 3 good
**Contribution:** 3 good
**Rating:** 5
**Confidence:** 3

**Summary:**

The paper presents an end-to-end pipeline for inpainting values of a diffusion process. The model is a hybrid of explicit (solutions to partial differential equations) and neural (U-net) components, where the evolution of the diffusion process is explicitly computed, but guided by learned parameters. The method is demonstrated on inpainting of optical flow fields, where it bests several chosen baselines that are explicit, neural, and neruoexplicit.

**Strengths:**

The particular combination of learned and explicit diffusion computation is novel.
The metrics demonstrate accuracy superior to the baselines.
The ablation study in Section 4.3 is informative.

**Weaknesses:**

The approach has only been demonstrated on one niche application -- optical flow. The paper does mention sparse mask inpainting of images several times, which could be another use case to strengthen the paper.  More results would be appreciated too, perhaps on some real-world datasets such as KITTI.

**Questions:**

Figure 1 would be more readable with larger fonts and more separation between the UNet and the D,a arrows. What is the difference between yellow and orange layers in the encoder? The inpainting boxes could be more fleshed out to visualize what they are actually doing (are they solving equation 8?). Where do the iterations come into play?

How does the diffusion tensor D connect to equation 8.

Section 3.1 mentions using average pooling to obtain the coarse version of the sparse flow field. Won't that grossly underestimate the flow field due to all the 0 values? Are those ignored somehow? Are the flow values also scaled down by 2 in each downsampling step, so that they are valid offsets for the coarser image size (similar for upsampling)?

Table 1 could be augmented with train/inference timings, parameter count, and number of iterations. The Figure 3 could be removed and that space used for additional results.

In Figure 2 left, it would be helpful to put the x axis ticks exactly where the samples are. There are only 4 sample sizes, and marking e.g. 0 on the x axis is really not informative.

In Figure 2 right, what does the vertical line down the middle indicate? Is that some ideal mask density threshold?

This sentence is hard to parse: "When evaluated on a density of 10%, the network trained on 5% density can even reach a very close EPE on to the network that was optimized on this density (0.28 vs. 0.29)." Does this intend to state that the network trained on 5% density has EPE of 0.29, while the network trained on 10% density has EPE of 0.28, when both are evaluated on 10% density dataset?

---

> ### Author Response · Authors · 2023-11-23
> **Part 1:**
>
> We sincerely thank the reviewer for the effort and appreciate the detailed, valuable feedback.
>
> **W1:** *"The approach has only been demonstrated on one niche application -- optical flow. The paper does mention sparse mask inpainting of images several times, which could be another use case to strengthen the paper. More results would be appreciated too, perhaps on some real-world datasets such as KITTI"*
>
> We agree that more results to demonstrate the effectiveness of our approach in a practical application strengthen the paper. The ground-truth optical flow in the KITTI dataset is acquired from the registration of LiDAR scans, and therefore inherently sparse in nature. Densifying it presents a practically highly relevant use case of our method. We have evaluated all our methods on this task, including the previous state of the art of optical flow inpainting from random masks. We provide the results in the new Section 4.4 and Table 3 in the paper. Our method is on- par with the Laplace-Beltrami method in terms of EPE but significantly outperforms most other methods in the FL metric - especially in the most difficult 1% density case - indicating that our results have fewer outliers. It demonstrates the impact of our method on this selected practical application. Please note that our method was not even adapted for the spatially varying mask densities present in this setting.
>
> General image data differs significantly from the flow field inpainting scenario that we designed our approach for. On one hand, flow fields are piecewise smooth, while natural images tend to contain texture and fine-scale details. While it is reasonable to use our approach for other piecewise smooth or cartoon-like data, significant changes would be required for natural image data. Moreover, the model-based inpainting in our hybrid approach uses the reference image to guide a linear diffusion process. This would not be available in a natural image inpainting scenario and thus would necessitate a different architecture. Here, a nonlinear process would be required which can infer structure directions from the given sparse data only. Overall, this would be a significant departure from the concepts we have proposed in our manuscript. We agree that addressing inpainting of images with neuroexplicit models is a highly interesting research question and will pursue this in future work.
>
> We thank the reviewer also for the additional valuable improvement suggestions and have addressed all of them in the paper:
>
>
> **Q1:** *"Figure 1 would be more readable with larger fonts and more separation between the UNet and the D,a arrows. What is the difference between yellow and orange layers in the encoder? The inpainting boxes could be more fleshed out to visualize what they are actually doing (are they solving equation 8?). Where do the iterations come into play?"*
>
> We adapted our figure 1 accordingly. Please see the updated version of the paper.
>
> **Q2:** *"How does the diffusion tensor D connect to equation 8"*
>
> In Equation, 6 we introduced the notation $D:=g(S)$, which states that the diffusion tensor is derived from the structure tensor $S$. In Equation 7, we use this notation to introduce the function
> $\Phi(I,~Ku^k)=g\Bigl(\sum_{i=0}^c(KI)_i (KI)_i^\top\Bigr)(Ku^k)$. The argument to the function $g$ here is a discrete version of the structure tensor $S$. Consequently, the diffusion tensor is inherently built into the function $\Phi$ which we reference in Equation 8. The full scheme in Equation 8 denotes one timestep (from k -> k+1) of the diffusion evolution we use to inpaint in the image-driven inpainting blocks in the Figure. Therefore, each of these blocks in the figure performs the number of timesteps we declare in Section 4.1. The first one performs 5 explicit steps, the next one 15, and so on.
>
> **Q3:** *"Section 3.1 mentions using average pooling to obtain the coarse version of the sparse flow field. Won't that grossly underestimate the flow field due to all the 0 values? Are those ignored somehow? Are the flow values also scaled down by 2 in each downsampling step, so that they are valid offsets for the coarser image size (similar for upsampling)?"*
>
> We thank the reviewer for pointing this out. Naive average pooling can indeed not be used due to the sparsity of the available data. We mentioned in the manuscript that we use averages of the flow values. This means that only flow values indicated by the binary mask are averaged. We do not scale the flow values at lower resolutions, since it does not matter if they are valid offsets or not. We are only concerned with validity on the full resolution, and therefore don't have to worry about scale on the lower resolutions. The coarse-to-fine process is purely done to speed up the inpainting process. We have addressed coarse-to-fine also in our answer to Reviewer 1, W3.

---

> ### Author Response · Authors · 2023-11-23
> **Part 2:**
>
> **Q4:** *"In Figure 2 left, it would be helpful to put the x axis ticks exactly where the samples are. There are only 4 sample sizes, and marking e.g. 0 on the x axis is really not informative.
> In Figure 2 right, what does the vertical line down the middle indicate? Is that some ideal mask density threshold?"*
>
> In the right part of Figure 2, we test how well all methods can deal with previously unseen mask densities. All methods were trained / optimized on a mask density of 5% (indicated by the gray line) and tested on different mask densities. This showed that the explicit and neuroexplicit methods have superior performance when presented with different mask distributions compared to the neural methods (see the increase in EPE of the neural methods when presented with more dense initializations compared to the one they were optimized on).
>
> We have adjusted the figures to make them more readable and space conserving in the manuscript.
>
>
> **Q5:** *"This sentence is hard to parse: "When evaluated on a density of 10%, the network trained on 5% density can even reach a very close EPE on to the network that was optimized on this density (0.28 vs. 0.29)." Does this intend to state that the network trained on 5% density has EPE of 0.29, while the network trained on 10% density has EPE of 0.28, when both are evaluated on 10% density dataset?"*
>
> This is correct, training the model on a specific density only provides a marginal performance benefit due to the good generalization to unknown mask densities of our method.

---

### Official Review · Reviewer_BStA · 2023-10-31

**Soundness:** 4 excellent
**Presentation:** 4 excellent
**Contribution:** 4 excellent
**Rating:** 6
**Confidence:** 4

**Summary:**

The paper presents a new approach that combines model-driven and data-driven methods to achieve improved inpainting of optical flow fields. The authors propose a joint architecture that integrates explicit partial differential equation (PDE)-based approaches with convolutional neural networks (CNNs). The paper demonstrates that their model outperforms both fully explicit and fully data-driven baselines in terms of reconstruction quality, robustness, and amount of required training data.

**Strengths:**

1. The paper successfully combines the strengths of explicit PDE-based models and CNNs, leveraging the interpretability and generalization capabilities of the former and the learning power of the latter. This integration provides an effective architecture for inpainting optical flow fields.

2. The proposed model achieves superior results compared to both explicit and data-driven baselines. The evaluation demonstrates higher reconstruction quality, robustness, and generalization capabilities, making it an advancement in the field.

2. The neuroexplicit diffusion model requires comparatively fewer learnable parameters and can be trained with significantly less data while still outperforming baselines trained on the full dataset. This aspect addresses the dependency on large-scale datasets, making the model more practical and efficient.

**Weaknesses:**

1. Although the paper compares the proposed model with explicit and data-driven baselines, it would be beneficial to include a comparison with other recent state-of-the-art methods in inpainting for optical flow fields. This would provide a more comprehensive evaluation and enhance the paper's contribution.

2. The paper assumes prior knowledge of diffusion processes and their application in inpainting. I wonder why diffusion-based inpainting is suitable for flow inpainting? Are there any theoretical explanations for this?
There are also many other traditional inpainting methods, are they also suitable in this task and do they work well with neural networks? Why or why not?

3. In the ablation study part, I wonder is coarse-to-fine approach important in this method? And is it possible to substitute Diffusion Tensor module with other parameter-free inpainting or propagation methods, to see which one best suits this task?

**Questions:**

Please address the questions in weakness part.

---

> ### Author Response · Authors · 2023-11-23
>
> We sincerely thank the reviewer for the effort and appreciate the detailed, valuable feedback.
>
> **W1:** *"It would be beneficial to include a comparison with other recent state-of-the-art methods in inpainting for optical flow fields"*
>
> We agree that comparing to the state-of-the-art baseline is useful in addition to previous comparisons against different methodologies. We have evaluated the state of the art. Our results show that we consistently outperform all competitors and thus set a new state of the art. Please see the section “Comparison against the state-of-the-art baseline” in the general comment to all reviewers and the updated table 1 in the paper for details.
>
> **W2:** *"I wonder why diffusion-based inpainting is suitable for flow inpainting"*
>
> In contrast to other model-based inpainting approaches such as the Absolutely Minimizing Lipschitz Extension (AMLE) and Laplace-Beltrami (LB) approaches, diffusion inpainting is our first choice due to several theoretical and practical considerations.
>
> First off, diffusion inpainting is closely related to regularization in model-based variational models for optical flow estimation. The so-called smoothness term in such cost functions is responsible for filling in flow data at regions of low confidence. This has a natural relation to diffusion filters, which is explained in (Weickert and Schnörr, 2001 [1]). Several variants of diffusion with edge preservation capabilities have been shown to perform well on the reconstruction of piecewise smooth image content from sparse data in compression (Schmaltz et al., 2014 [2]), including flow fields (Jost et al. 2019 [3]). Edge-enhancing diffusion represents the state of the art among this class of diffusion filters.
>
> Another advantage of diffusion compared to AMLE or LB is the existence of theoretical results w.r.t. combinations with deep learning. For instance, for diffusion filters as layers in neural networks, provable stability guarantees have been established (Alt et al. 2022 [4]).
>
> The aforementioned advantages make diffusion inpainting a natural choice for the model-based part of our neuroexplicit model. The experimental results in Table 1 also confirm our results experimentally: Our model outperforms all neural and learning-based competitors. . Nevertheless, other approaches such as AMLE or LB are interesting candidates for additional future research regarding neuroexplict flow inpainting.
>
>
>
> **W3:** *"Is coarse-to-fine approach important in this method"*
>
> The coarse-to-fine approach is critical to speed up the convergence rate of the method, which is determined by the distance across which information has to be transported. Working on the full resolution would require a high number of iterations of the diffusion model and make the optimization within the deep learning framework intractable.
> Inpainting results from coarse levels provide a good initialization for finer resolutions of the inpainting problem. This allows to speed up convergence on the next resolution level in a straightforward and efficient way without impacting the quality of the reconstruction. The results from Table 1 confirm that there is no qualitative disadvantage due to the coarse-to-fine scheme, and it enables us to train our models and achieve state of the art.
>
> **References:**
>
> [1] Joachim Weickert and Christoph Schnörr. A theoretical framework for convex regularizers in pde-
> based computation of image motion. International Journal of Computer Vision, 45(3):245–264,
> December 2001. ISSN 0920-5691
>
> [2] Christian Schmaltz, Pascal Peter, Markus Mainberger, Franziska Ebel, Joachim Weickert, and An-
> drés Bruhn. Understanding, optimising, and extending data compression with anisotropic diffu-
> sion. International Journal of Computer Vision, 108(3):222–240, jul 2014. ISSN 0920-5691.
>
> [3] Ferdinand Jost, Pascal Peter, and Joachim Weickert. Compressing flow fields with edge-aware
> homogeneous diffusion inpainting. In Proc. 45th International Conference on Acoustics, Speech,
> and Signal Processing, pp. 2198–2202, Barcelona, Spain, May 2020. IEEE Computer Society
> Press.
>
> [4] Tobias Alt, Karl Schrader, Matthias Augustin, Pascal Peter, and Joachim Weickert. Connections
> between numerical algorithms for pdes and neural networks. Journal of Mathematical Imaging
> and Vision, 65(1):185–208, jun 2022. ISSN 0924-9907

---

### Author Response · Authors · 2023-11-23
**General comment to address all reviewers**

All reviewers suggested our manuscript could be strengthened further by more experiments. We thank the reviewers for their effort and the valuable feedback. In the following, we would like to provide the requested evaluations.

**Comparison against the state-of-the-art baseline:**

We thank reviewer 3 for the reference to the work “On Anisotropic Optical Flow Inpainting Algorithms” from Raad et al. [1]. Using Google Scholar we have verified that this work is indeed the most recent method addressing our task of optical flow inpainting with random masks and thus constitutes the prior state of the art. We evaluated against their publicly available code using their hyperparameters that were already tuned for the target dataset Sintel. Note that our method was trained on FlyingThings and was not tuned for Sintel. Even with this disadvantage, our method consistently outperforms both of their variants by 23.76% and 10.97% on the Sintel dataset. With these results, our method sets a new state of the art for the given task. Please see the updated Table 1 in the paper for details.

**Evaluation on the KITTI dataset for autonomous driving:**

The ground truth optical flow in the KITTI dataset is acquired from the registration of LiDAR scans, and therefore inherently sparse in its nature. Densifying it presents a practically highly relevant use case of our method. In the updated supplemental material, we evaluate all trained models and explicit methods on the KITTI datasets.  To this end, we subsampled the already sparse data and measured how well the left-out samples can be reconstructed by our inpainting method and all baselines - except probabilistic diffusion - as this baseline cannot be executed on the different image resolution. Notably, none of the methods was tuned for KITTI. Our method is on par with the Laplace-Beltrami method in terms of EPE but significantly outperforms most other methods in the FL metric - especially in the most difficult 1% density case - indicating that our results have fewer outliers. It demonstrates the positive impact of our method on this selected practical application. Please see the new Section 4.4 and Table 3 in the paper for details. Please note that our method was not even adapted for the spatially varying mask densities present in this setting.


**References**:

[1] Lara Raad, Maria Oliver, Coloma Ballester, Gloria Haro, and Enric Meinhardt. On anisotropic
optical flow inpainting algorithms. Image Processing On Line, 10:78–104, 2020

---

### Meta-Review · Area_Chair_1WbT · 2023-12-05

**Metareview:**

The paper suggests combining multiple explicit PDE approaches which CNNs in order to obtain the "best of both worlds" (the explainability of the PDE and the accuracy of the CNNs). The paper shows that the joint architecture improves on inpainting optical flow fields for Sintel but does not compare against other techniques on either KITTI flow or other vision tasks. I agree with Reviewers 3rtj and cyeS that more work needs to be done to show that the technique is effective in general.

**Justification For Why Not Higher Score:**

I agree with Reviewers 3rtj and cyeS that more work needs to be done to show that the technique is effective in general.

**Justification For Why Not Lower Score:**

N/A

---

### Decision · Program_Chairs · 2024-01-16

Reject